# Synergistic Effect of Ultrasound and Polyethylene Glycol on the Mechanism of the Controlled Drug Release from Polylactide Matrices

**DOI:** 10.3390/polym11050880

**Published:** 2019-05-14

**Authors:** Wenting Bao, Xianlong Zhang, Hong Wu, Rong Chen, Shaoyun Guo

**Affiliations:** 1The State Key Laboratory of Polymer Materials Engineering, Polymer Research Institute of Sichuan University, Chengdu 610065, China; baowenting@fccc.org.cn (W.B.); wh@scu.edu.cn (H.W.); rongchen@scu.edu.cn (R.C.); nic7702@scu.edu.cn (S.G.); 2Aviation Fuel & Chemical Airworthiness Certification Center of CAAC, The Second Research Institute of Civil Aviation Administration of China, Chengdu 610207, China

**Keywords:** ultrasound, polylactide, polyethylene glycol, drug release, hot melt extrusion

## Abstract

In this paper, the synergistic effect of ultrasound and polyethylene glycol (PEG) on the controlled release of a water soluble drug from polylactide (PLA) matrices was studied. When ultrasound was used following the hot melt extrusion (HME) of the PLA/model drug release system, the release of the model drug (Methylene Blue (MB)) from the PLA when immersed in phosphate buffered saline (PBS) was affected by the variation of the parameters of ultrasound. It was found that no more than 2% PLA dissolved during the in-vitro release study, and the release of the MB from the PLA was diffusion controlled and fit well with the Higuchi diffusion model. Polyethylene glycol (PEG), which has high hydrophilicity and rapid dissolution speed, was blended with the PLA during the melt extrusion to enhance the release of the MB. The analysis of the structure and properties of the in-vitro release tablets of PLA/PEG/MB indicated that the ultrasound could improve the dispersion of MB in the PLA/PEG blends and it could also change the structure and properties of the PLA/PEG blends. Due to the dissolution of the PEG in PBS, the release of the MB from the PLA/PEG drug carrier was a combination of diffusion and erosion controlled release. Thus a new mechanism combining of diffusion and erosion models and modified kinetics model was proposed to explain the release behavior.

## 1. Introduction

The development of controlled release products is one of the important applications of biodegradable and bioresorbable polymers [1,2,3,4,5]. A variety of polymers have been used for such applications [6,7,8,9]. Polylactide (PLA) is one of the most extensively investigated polymers for implantable or injectable drug delivery systems. Several physical forms have been studied, including implants, microparticles, and nanoparticles [10,11,12,13]. The biocompatibility of PLA has been demonstrated by its long clinical use in physiological environments as they are hydrolyzed into metabolic products which are eliminated from the body [14].

The transport of drugs from delivery systems based on polymers is a rather complex process. The drug delivery behavior is affected by many factors, including polymer molecular weight, copolymer ratio, composition of blends, polymer crystallinity, preparation method, properties of the incorporated drug, etc. [15,16,17,18,19,20,21,22,23,24,25,26]. As reported in the literature, most of the PLA based drug delivery systems have been prepared by traditional methods involving solvent processes. However they have showed various problems, such as environmental pollution and residual organic solvent [27]. PLA, which is one of the thermoplastic polymers, becomes soft and flowable when heated above its melting point and can be shaped into a variety of products by several molding techniques, such as injection, compression, and extrusion. In recent years, hot melt extrusion (HME), similar to normal melt extrusion as used for non-pharmaceutical purposes, has been introduced into the field of pharmaceutical preparation [28,29,30,31,32]. Preparing PLA based drug delivery system by extrusion has many advantages [31,32]. For instance, blending by melt extrusion has been widely used for polymer modification. The release properties of PLA based drug carriers should be controllable by blending PLA with other components via extrusion. However, the limited stability of PLA at high temperatures and especially under shearing forces [33,34], and the poor heat stability of most drugs [31] have been the main limitation and concern when using melt extrusion processing. In our previous studies, ultrasound was applied to assist melt processing and we showed that the ultrasound could break the polymer chains into small fragments and improve the dispersion of functional fillers [35,36,37,38,39,40,41,42]. However, so far, to our knowledge, there have been only a few reports on the release behavior of PLA based drug delivery system prepared by melt processing. It was found that ultrasound could improve the release and solubility of a drug in water [43,44,45].

Therefore, our goal for the research described here was to develop a PLA based long term drug delivery system and control the release behavior by melt blending and subsequent ultrasound treatment. Methylene blue (MB) was used as the model drug [46,47] since it is stable under extrusion processing. The PLA based drug delivery system was prepared by melt extrusion. The effect of subsequent ultrasound melt treatment and blending hydrophilic polyethylene glycol (PEG) with the PLA in the extrusion process on the release behavior of model drug was studied.

## 2. Experimental

### 2.1. Materials and Equipment

The PLA (REVODE101, Mn = 5.3 × 10^4^, Mw/Mn = 1.935) used in the experiment was provided by Zhejiang Hisun Biomaterials Co., Ltd. China. It is a thermoplastic resin derived from annually renewable resources and specifically designed for extrusion applications. The melt flow index and density of the PLA were 10 g/10 min and 1.25 ± 0.05 g/cm^3^, respectively. The PEG with molecular weight of 6000 used was provided by Tianjin Bodi Chemical Co., Ltd., Tianjin, China. The molecular weight of the model drug MB was 373.90 and was also provided by Tianjin Bodi Chemical Co., Ltd., Tianjin, China.

The ultrasound treatment was carried out on the extruded granules by a special ultrasound applicator constructed in our laboratory as shown in Figure 1; the diameter of the probe was 15 mm, maximum power output was 250 W, and frequency was 20 kHz.

### 2.2. Sample Preparation

The PLA and PEG were first dried in a vacuum oven at 40 °C for 24 h. Selected amounts of PLA, PEG and MB (Table 1) were mixed in a sealed bag and then the mixture was added into an extruder, Thermo Scientific HAAKE MiniLab, Thermo Scientific, Waltham, MA, USA. The temperature was set at 170 °C and the screw rotation speed was fixed at 50 rpm. After 3 min cycle mixing in the extruder, the PLA/PEG/MB blends were extruded and cut into granules. 

1.5 g dried PLA/MB or PLA/PEG/MB granules were filled into the cell shown in Figure 1 and kept at 170 °C for 5min, then the ultrasound irradiation with different intensities was induced to the melts for 0 or 3 min, held at 170 °C for an additional 5 or 2 min to make sure that the total treating time at 170 °C was 10min. The treated PLA/MB or PLA/PEG/MB melts were then cooled down in the cell naturally to room temperature. Table 2 lists the ultrasound processing parameters.

After the above steps, the prepared blends of PLA/MB and PLA/PEG/MB were vacuum dried at 40 °C for 24 h, cut into small pieces, and then compression-molded on a Compression Molding Machine (HP-63, Xima Weili Plastic Machine Factory, Guangzhou, China) at 170 °C under 10 MPa for 3 min; cooling was accomplished by water through the platens at 25 °C under 10 MPa for 10 min. Small tablets were obtained, with the diameter of 10mm and the thickness of 0.4mm, to be used for in the in-vitro release studies.

### 2.3. Measurements

#### 2.3.1. In-Vitro Release Study

In-vitro drug release from the PLA and PLA/PEG matrices was measured for up to 2880 h (120 days) considered for long term drug release application, about 35 mg of tablets incubated in 5 mL phosphate buffered saline (PBS, ZLI-9062, Zhongshan Golden Bridge Biotechnology Co., zhongshan, China) at 37 ± 1 °C in an incubator (HZQ-X100, Haocheng Co., Changzhou, China) with the PBS stirred at 100 rpm. Test tubes that contained the tablets and 5ml PBS (pH = 7.4) were placed in a shaker at 37 ± 1 °C. At each predetermined times varying from 4 to 2880 h (120 days), 5 mL medium was taken out and 5 mL of new PBS solution was added. Then the concentration of MB in the removed medium was measured by an ultraviolet-visible spectrophotometer (UV-1750, Shimadzu Corp., Kyoto, Japan) and calculated based on the intensity of the max absorption peak at 664 nm [48,49].

#### 2.3.2. Weight Loss

Before the in-vitro release tests, the tablets were dried to a constant weight in a vacuum oven at 37 °C; the weight of the tablets before the release tests were named as W_0_. When the in-vitro release tests were finished at 120 days, the tablets were removed from the tubes and washed with deionized water 3 times, then the samples were again dried to a constant weight in a vacuum oven at 37 °C; the weight of the tablets after the release tests were named as W_1_. Therefore, the weight loss of the tablets during the in-vitro release tests could be calculated, as defined by Equation (1).
(1)$$\mathrm{Weight}\;\mathrm{loss}(\%)=\frac{W_{0}-W_{1}}{W_{0}}\times 100\%$$

#### 2.3.3. Scanning Electron Microscopy (SEM)

After the in-vitro release tests, the tablets were cryogenically fractured in liquid nitrogen. Then the morphologies of the molded surface and the fracture cross section of the PLA/PEG/MB tablets after coating with gold for conductivity, were studied by scanning electron microscopy (SEM, JSM-5900LV, JEOL Ltd., Tokyo, Japan).

## 3. Results and Discussion

### 3.1. Effect of Ultrasound on the In-Vitro Release Behavior of MB from PLA

Figure 2 shows the release curves of MB from the PLA as a function of power of the ultrasound treatment. In the first 20 days, the release rates of MB from the tablets treated by ultrasound were faster than that from the tablets without ultrasound treatment. From 20 to 120 days, all of the release rates became smaller. From about 30 to 120 days, when the ultrasonic power was 50 or 250 W, the cumulative releases of MB were less than that from PLA without ultrasound treatment. However, when the ultrasonic power was 150 W, both the release rate and the cumulative release of MB from PLA were improved, with the amount being longer then both the non-treated PLA and the 50 and 250 W treated samples for the entire period of measurement.

We suggest the main reason for this phenomenon was that the ultrasound had two different kinds of effects on the release of MB from PLA: firstly, ultrasound would lead to the degradation of the PLA matrix resulting in more PLA chains with lower molecular weight in the tablets [40], there would be more hydroxyl group, so the degradation and hydrophilicity of PLA would be improved and this kind of effect would enhance the release of the MB. Secondly, ultrasound would improve the dispersion of MB in the PLA matrix, there was more MB dispersed in the internal part of the tablet, which would take longer to diffuse into the medium, so this effect would decrease the release of MB. These two effects competed with each other and led to the final results. In the first stage of the immersion, the degradation effect of ultrasound was predominant. After 20 days immersion, the effect of the polymer matrix on the release became weak and the distribution and diffusion of the drug was the dominant effect. We assume that when the ultrasound was applied at 150 W for 3 min, these two effects achieved a balance, so the molecular weight of PLA was decreased obviously and the dispersion of the drug MB was improved slightly, which led to the increase of both the release rate and cumulative release of MB for the entire measurement period.

Table 3 lists the data of the weight loss of PLA/MB tablets after 120 days immersion in the PBS medium. The weight loss was very small and there was no obvious corrosion of the PLA matrix, as shown by Figure 3a,a1, occurring during the release process. Thus, we concluded that the release of MB from the PLA matrix was diffusion controlled.

### 3.2. Effect of PEG on the In-Vitro Release Behavior of MB from PLA, All without Any Ultrasound Treatment

MB is water-soluble and its release behavior would be affected by the properties of the polymer matrices. The release curves of MB from PLA and the PLA/PEG blends without any ultrasound or heat treatment are compared in Figure 4. From this figure, it could be observed that the release of MB from pure PLA was very slow and after 120 days immersion there was only 10.5% of the initial MB released from the tablets. Through blending PLA with PEG, the release of MB from the PLA/PEG matrices increased to about 70% for all those samples with PEG. We suggested that the model drug MB was dispersed predominantly in the PEG phase, since the PEG had lower molecular weight and better hydrophilicity than PLA. 

With the increase of the content of PEG, the initial release rate of MB increased. The release rates of MB from the tablets to which 20% PEG and 35% PEG were added were lower than that of the tablets with 50% PEG and decreased slowly in the first 60 days. In the first two days, the tablets to which 50% PEG were added showed the highest release rate of MB and then the release rate decreased sharply since 70% of the MB had been released. These different release behaviors could be attributed to the different compositions of the polymer matrices. In the cases of PLA and its blends with PEG, since PEG is a hydrophilic and non-toxic biomaterial which has been usually used to improve the hydrophilicity of aliphatic polyesters, it was known that the hydrophilicity of the polymer matrix decreased in the order of PLA/PEG (50/50) > PLA/PEG (65/35) > PLA/PEG (80/20) > PLA [50,51]. Simultaneously, the degradation rate of the polymer matrix decreased in the same order. Therefore, with better hydrophilicity and faster degradation rate of the polymer matrix, the release rate of MB was fastest for the PLA/PEG 50/50 blend. 

The data of the weight loss of the tablets after 120 days are listed in Table 4. The weight loss of PLA was only 0.36%, while the weight losses of the PLA blended with PEG were about 18.8%, 28.5%, and 41.0%, each of which nearly reached their theoretical PEG content. This meant that pure PLA degraded very slowly and the weight loss should be attributed to the PEG dissolving from the tablets into the PBS medium. 

Figure 3 shows the SEM photomicrographs of the cryofacture cross sections and the surfaces of the PLA/PEG/MB tablets after 120 days immersion in PBS all at the same magnification. It could be seen clearly that there were many more pores on the surfaces of the tablets than on the cross sections. This result indicated that the dissolution of the PEG of the polymer matrices and the dissolution of MB from the surface were higher than those from the internal part of the tablet and it also indicated that there was more PEG initially dispersed on the surface of the tablets.

The porous structure, which should be attributed to the dissolution of polymer matrices and the dissolution of MB, was affected by the compositions of the polymer matrices. Due to the slow degradation rate in PBS and poor hydrophilicity of pure PLA, the release of MB from pure PLA was very slow and there were only very tiny and thin pores. With the addition of PEG, the degradation rate in PBS and the hydrophilicity of the polymer matrices were enhanced with the rate increasing with increasing PEG content. Therefore, the dimension and number of the pores increased with the increase of the PEG content. These results were consistent with the in-vitro release experiments. 

### 3.3. Effect of Ultrasound and PEG on the In-Vitro Release Behavior of MB from PLA

Based on the above results, both ultrasound and blending PEG with PLA were applied to study their synergistic effect on the release behavior of MB from PLA. As shown in Figure 4, the release of MB was increased by blending PLA with 35% PEG as compared to the pure PLA. Figure 5 shows the release curves of MB from PLA blended with 35% PEG with various ultrasound treatments. In the first 4 days, all of the tablets treated by ultrasound showed higher release rates of MB than those without ultrasound treatment (PLA/PEG/MB H10), and subsequently, the release rate of MB from the tablets treated by 50W ultrasound became the slowest. When the ultrasound was applied at 150 W for 3 min, the release rate and cumulative released content of MB was the highest, and continued so for the 120 days.

As discussed above, ultrasound could control the release of MB from PLA through its effects on the degradation of the PLA matrices and the dispersion of MB in the tablets. When PEG was added, the carrier of the MB in the tablet became the PLA/PEG blends instead of PLA, and PLA/PEG blends were degraded by the chemical effect of ultrasound. This led to a faster hydrolysis degradation rate of the PLA/PEG matrices and a faster release rate of MB. At the same time, the physical effect of ultrasound might improve the dispersion of PEG and MB in PLA. The better dispersion would have two results. One was that the more and better dispersed release channels, which were formed by the erosion of PEG, would increase the release of the drug. The other was that the better dispersion of MB meant relatively more MB dispersed in the PLA phase, so the drug release became more difficult. When the ultrasound was applied at 150 W for 3 min, these two kinds of results achieved balance, so the release of MB from the tablets was the fastest and the most. 

Table 5 lists the data of the weight loss of the tablets after 120 days immersion. The differences of the weight loss of the PLA/PEG/MB tablets treated by different intensities of ultrasound were not obvious. Based on the previous discussion, the weight loss should be mainly attributed to the erosion of PEG and the release of MB. Comparing the initial weight of PEG with that of MB, it could be seen that PEG was dominant in the weight loss. Since the content of PEG in these tablets was the same, the weight loss of the tablets that were treated by different intensities of ultrasound were similar. 

The cross sections and surfaces of the PLA/PEG/MB tablets after 120 days immersion are shown in Figure 6. In this figure, the pores on the surfaces of the tablets were larger than those on the cross sections, which was same as the results shown in Figure 3. In addition, the dimension of the pores on the surface increased with the increase of ultrasound intensity. When the ultrasonic intensity was 150 W, the number of the pores on the cross section was the largest, although smaller than for 250 W. So it could be speculated that the tablets treated by 150 W ultrasound had more release channels through the erosion of PEG and more MB released. This result was consistent with the data of the in-vitro release test.

### 3.4. Mechanism and Kinetics of the Release of MB from PLA Based Matrices

MB is a kind of water-soluble material, but there was no obvious weight loss in the in-vitro release from the PLA/MB tablets (as shown in Table 3, the weight loss was smaller than (1.2 ± 0.2) %), it could be inferred that the release of the drug from PLA was diffusion controlled and the diffusion channels included the tiny pores from the degradation of PLA and the pores left by the release of the drug.

By blending PEG with PLA, without ultrasound or heat treatment, the release of MB from PLA/PEG was ascribed to not only the erosion of PEG, but also the diffusion of MB from PLA. So the release of MB from PLA/PEG could be divided into two steps: at the beginning of the release, the release rate of MB through the erosion of PEG was higher than that through the diffusion from the PLA phase, so the first stage was an erosion controlled stage; after several days release, most of the PEG was dissolved in the medium and a lot of pores were left in the tablet and the release of MB was controlled by its diffusion from the porous PLA matrix, so the second stage was diffusion controlled. 

From the all of the above results and discussions of the results of ultrasound treated samples, we concluded that the physical and chemical effects of ultrasound on the release of MB from the PLA based matrices were shown as follows. First, PLA and PEG were degraded under ultrasound and the molecular weights of both polymers were decreased [40]. The PLA and PEG with lower molecular weights showed faster hydrolysis rates in PBS, so the drug release could speed up. Second, the ultrasound could affect the dispersion of the PEG phase in the PLA and this would affect the number and the dispersion of the diffusion channels formed by the erosion of the PEG. From the SEM figures it was found that the content of PEG on the surface of the tablet increased and the diameter of the dispersed phase PEG became smaller with the application of ultrasound. Third, since the model drug MB would normally tend to disperse more in the PEG phase than in the PLA phase, the ultrasound could affect the dispersion of the model drug MB in the two phases. And the hydrophilicity of PEG was better than PLA, the relatively more MB was in the PLA phase, the more difficult it was to be released from the tablet.

Therefore, after the application of ultrasound, the release behavior of the model drug MB from the PLA based matrices could be expressed as shown schematically in Figure 7. Based on the above discussion about the effect of ultrasound on the release of MB from the PLA based matrices and its mechanism, the release behavior of MB in PLA/PEG could be divided into two stages. During the erosion controlled stage, the release of MB was greatly influenced by the erosion of PEG; we suggest a first order kinetics model could be used to fit the release of MB at this stage since the first-order kinetic exponential model has usually been used for the dissolution of a solid particle in a liquid media [52,53]. In the diffusion controlled stage, the Higuchi model could be applied to model the kinetics of the release of MB from the PLA [54]. Since the release of MB came from both the erosion of the PEG phase and the diffusion from the PLA matrices, the cumulative release of the model drug MB could be expressed by the following equation (Equation (2)), where M_t_ is the amount of drug released in time t, M_0_ is the initial amount of drug in the tablet and Q is the amount of the drug released by the erosion of PEG. In this equation, k_1_ is the erosion release constant and is related with the content and dispersion of the PEG, and k_2_ is the diffusion release constant which is affected by the porous structure of PLA. Thus the first term on the right side corresponds to a first order kinetic and the second term is the Higuchi model.
(2)$$\frac{M_{t}}{M_{0}}=Q(1-e^{-k_{1}t})+k_{2}t^{\frac{1}{2}}$$

This equation was applied to fit the release of MB from PLA/PEG blends with the ultrasound treatment. The fitting curves are shown in Figure 8 and the kinetics parameters are listed in Table 6. The value of Q and the erosion release constant k_1_ were increased by the ultrasound treatment. When the ultrasound was applied at 150 W for 3 min, the value of Q and the erosion release constant were the maximum, which meant that the model drug released by the erosion of PEG was the most and fastest. The more model drug released due to the erosion of PEG, the smaller was the value of the diffusion release constant k_2_; this was because there was less drug left in the porous PLA in the latter stage of the release. Therefore, the data of the kinetics parameters were consistent with the above experimental results and discussion.

## 4. Conclusions

In this paper, the synergistic effect of ultrasound and PEG on the release of a model drug from PLA matrices was studied. By blending PLA with PEG, the release rate of the model drug, MB, from PLA could be increased significantly. The experimental results showed that ultrasound treatment of the PLA/PEG/MB release system could affect the dispersion of the model drug in the matrices and it could also affect the properties of the matrices. A new mechanism combining diffusion and erosion models was proposed to explain this release behavior and the release curves fit well with the proposed equation based on this mechanism. 

## Figures and Tables

**Figure 1 polymers-11-00880-f001:**
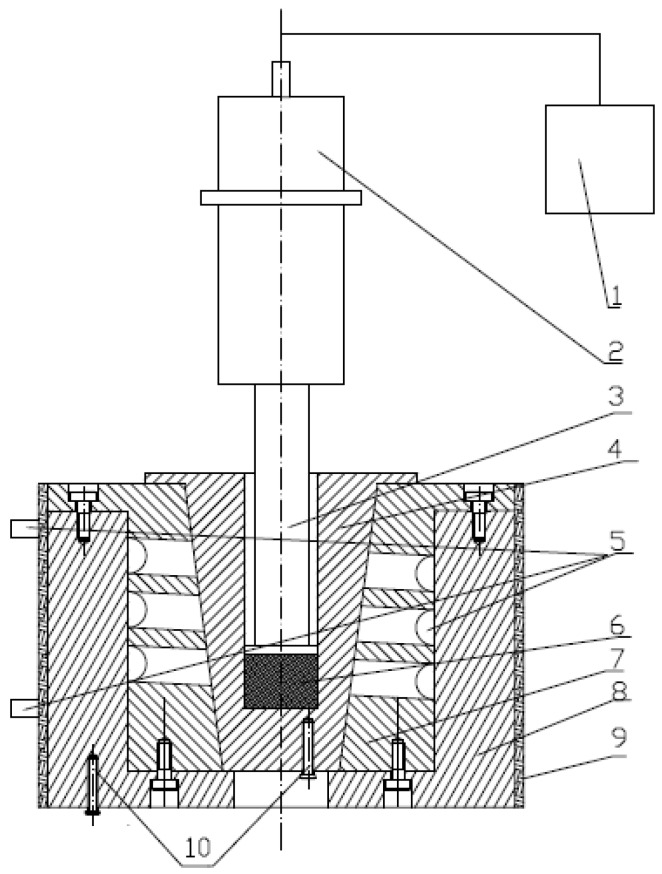
Scheme of the special ultrasound applicator. 1: Ultrasonic generator; 2: Piezoelectric transducer; 3: Probe; 4: Cell; 5: Cooling system; 6: Melt; 7, 8: Mold cavity; 9: Electric heater; and 10: Thermocouples.

**Figure 2 polymers-11-00880-f002:**
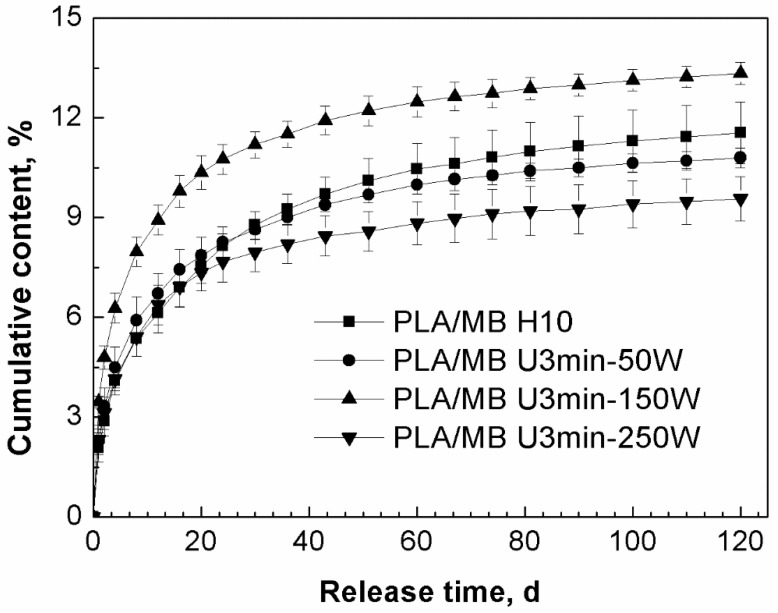
Effect of ultrasound on the release behavior of MB from PLA by immersion in phosphate buffered saline (PBS) for up to 120 days (PLA/ MB H10: without ultrasound treatment, PLA/MB U3min-50W: treated by 50 W ultrasound for 3 min, PLA/MB U3 min-150 W: treated by 150 W ultrasound for 3 min, PLA/MB U3min-250W: treated by 250 W ultrasound for 3 min).

**Figure 3 polymers-11-00880-f003:**
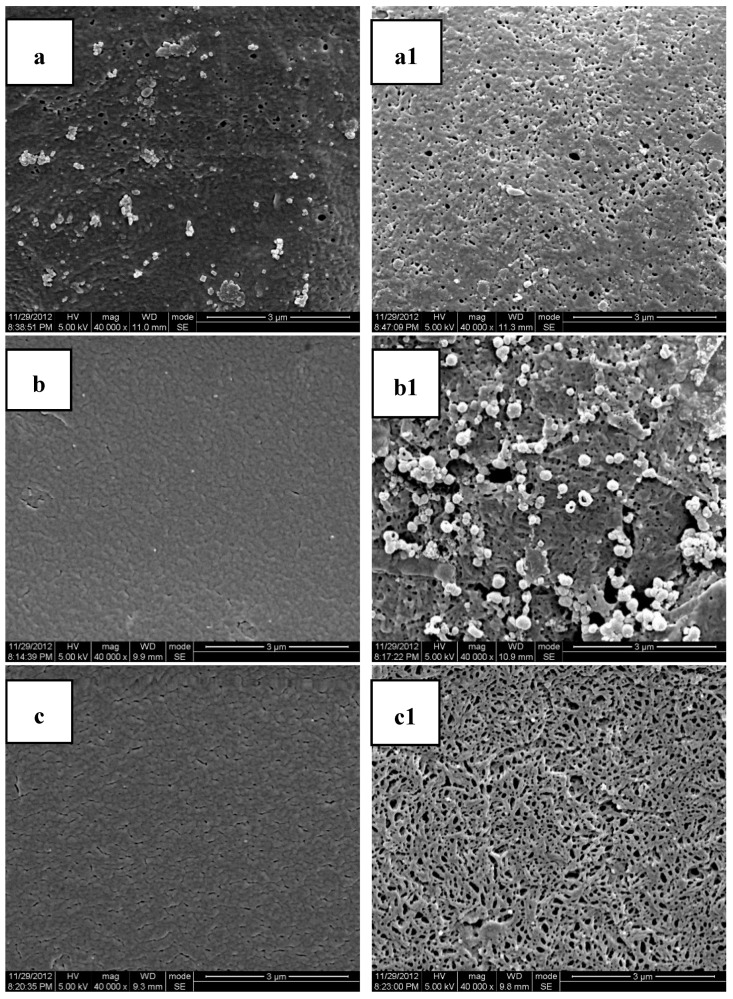
Scanning Electron Microscopy (SEM) photos of the cross sections and molded surfaces of the PLA/PEG/MB samples after 120 days immersion in PBS ((**a**,**a1**) PLA; (**b**,**b1**) PLA-20%PEG6000; (**c**,**c1**) PLA-35%PEG6000; (**d**,**d1**) PLA-50%PEG6000; (**a**–**d**) cross section; (**a1**–**d1**) molded surface).

**Figure 4 polymers-11-00880-f004:**
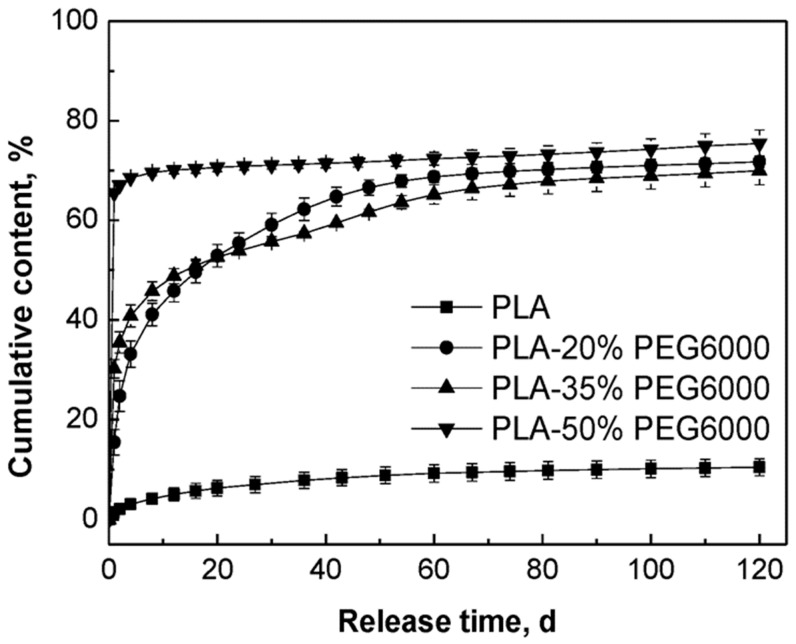
Effect of content of PEG on the in-vitro release behavior of MB from the PLA/PEG blends without ultrasound treatment.

**Figure 5 polymers-11-00880-f005:**
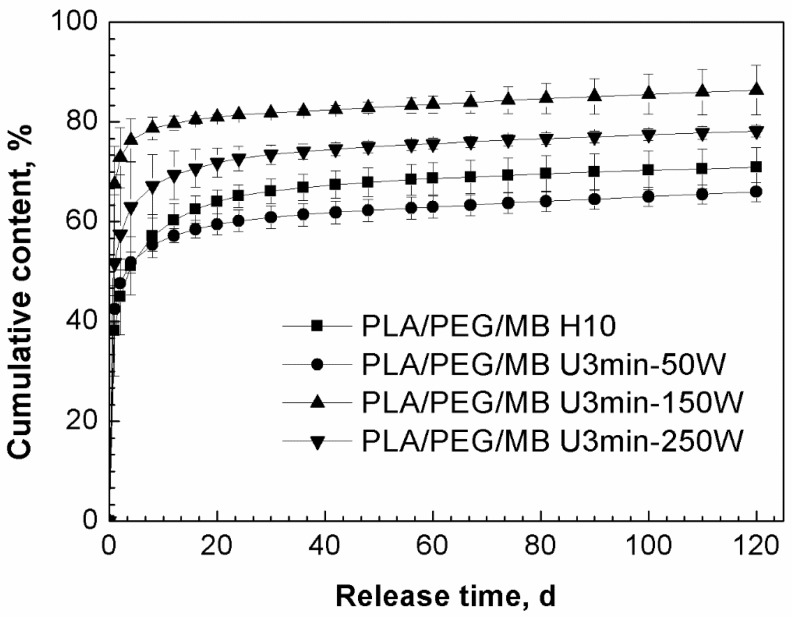
Effect of ultrasound on the release behavior of MB from PLA/PEG (65/35) blends immersed in PBS for 120 days (PLA/PEG/MB H10: without ultrasound treatment, PLA/PEG/MB U3min-50W: treated by 50 W ultrasound for 3 min, PLA/PEG/MB U3 min-150 W: treated by 150 W ultrasound for 3 min, PLA/PEG/MB U3min-250W: treated by 250 W ultrasound for 3 min).

**Figure 6 polymers-11-00880-f006:**
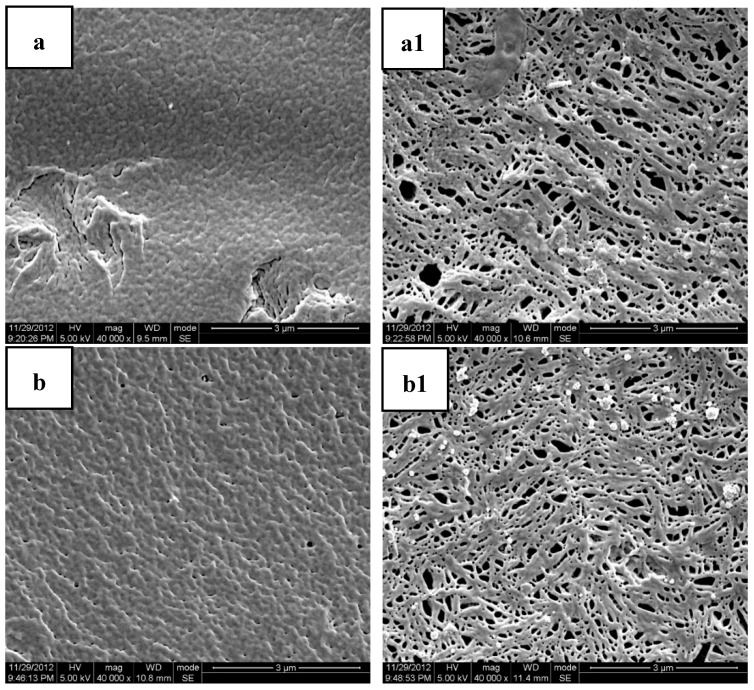
SEM photos of the cryofractured cross sections and molded surfaces of PLA/PEG/MB samples after immersion in PBS for 120 days ((**a**,**a1**) PLA/PEG/MB H10; (**b**,**b1**) PLA/PEG/MB U3min-50W; (**c**,**c1**) PLA/PEG/MB U3 min-150 W; (**d**,**d1**) PLA/PEG/MB U3min-250W.; (**a**–**d**) cross section; (**a1**–**d1**) molded surface).

**Figure 7 polymers-11-00880-f007:**
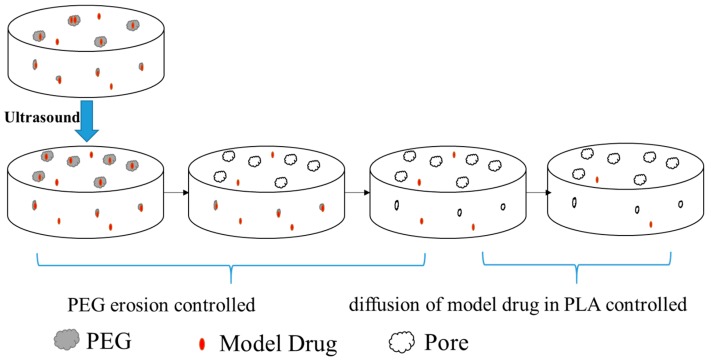
Scheme of the effect of ultrasound on the release of model drug MB from the PLA/PEG blends.

**Figure 8 polymers-11-00880-f008:**
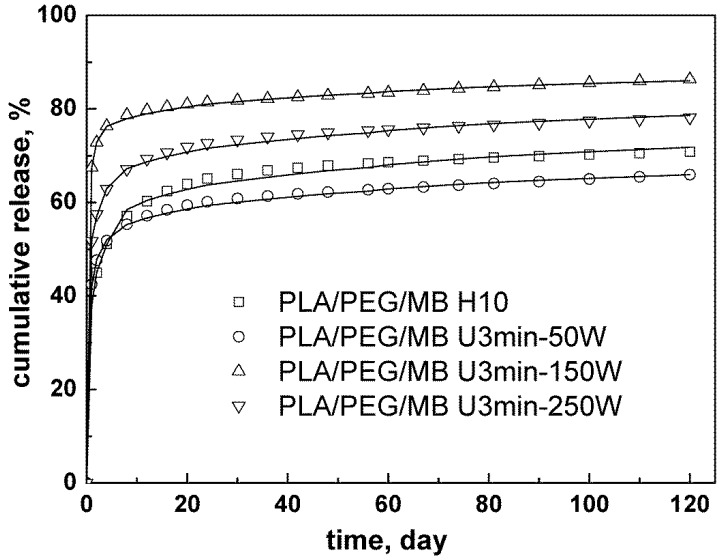
Kinetics analysis for the release of MB from PLA/PEG (65/35) blends with different ultrasound intensity.

**Table 1 polymers-11-00880-t001:** Composition of Polylactide (PLA)/polyethylene glycol (PEG)/methylene blue (MB).

Sample Name	Content of PLA, %	Content of PEG, %	Content of MB, %
PLA	100	0	1
PLA-20%PEG6000	80	20	1
PLA-35%PEG6000	65	35	1
PLA-50%PEG6000	50	50	1

**Table 2 polymers-11-00880-t002:** Processing parameters for PLA/MB (100/1) and PLA/PEG/MB(65/35/1).

Sample Name	Ultrasound Power, W	Preheating Time, min	Ultrasound Time, min	After-Heating Time, min
**PLA/MB** H10	0	5	0	5
**PLA/MB** U3min-50W	50	5	3	2
**PLA/MB** U3min-150W	150	5	3	2
**PLA/MB** U3min-250W	250	5	3	2
**PLA/PEG/MB** H10	0	5	0	5
**PLA/PEG/MB** U3min-50W	50	5	3	2
**PLA/PEG/MB** U3min-150W	150	5	3	2
**PLA/PEG/MB** U3min-250W	250	5	3	2

**Table 3 polymers-11-00880-t003:** Weight loss of the PLA/MB tablets after 120 days immersion in PBS.

Specimen	Weight of Sample (before Immersion), mg	Weight of Sample (after Immersion), mg	Weight Loss, %
**PLA/MB** H10	37.07 ± 0.39	36.92 ± 0.33	0.43 ± 0.18
**PLA/MB** U 3min-50W	38.19 ± 0.35	37.94 ± 0.42	0.65 ± 0.19
**PLA/MB** U 3min-150W	37.39 ± 1.35	37.28 ± 1.32	0.29 ± 0.10
**PLA/MB** U 3min-250W	38.07 ± 0.72	37.60 ± 0.82	1.21 ± 0.24

**Table 4 polymers-11-00880-t004:** Weight loss of PLA/PEG/MB tablets after 120 days immersion.

Sample Name	Weight of Tablet (before Immersion), mg	Weight of Tablet (after Immersion), mg	Weight Loss, %
PLA	36.42 ± 0.36	36.29 ± 0.42	0.36 ± 0.18
PLA-20%PEG6000	39.17 ± 0.71	31.80 ± 0.68	18.82 ± 0.15
PLA-35%PEG6000	37.87 ± 1.27	27.10 ± 1.01	28.47 ± 0.75
PLA-50%PEG6000	37.07 ± 0.24	21.80 ± 0.38	40.95 ± 0.08

**Table 5 polymers-11-00880-t005:** Weight loss of PLA/PEG/MB (65/35/1) tablets after 120 days immersion.

Sample Name	Weight of Sample (before Immersion), mg	Weight of Sample (after Immersion), mg	Weight Loss, %
**PLA/PEG/MB** H10	35.97 ± 0.63	25.52 ± 1.15	29.05 ± 4.66
**PLA/PEG/MB** U3min-50W	35.98 ± 1.32	25.45 ± 0.86	29.27 ± 1.44
**PLA/PEG/MB** U3min-150W	34.88 ± 0.89	25.09 ± 0.99	28.58 ± 2.34
**PLA/PEG/MB** U3min-250W	35.56 ± 0.79	24.7 6± 1.58	30.37 ± 2.61

**Table 6 polymers-11-00880-t006:** Effect of ultrasound on kinetics parameter, *Q*, erosion release constant, *k_1_*, and diffusion release constant, *k_2_*, and correlation coefficients, *R^2^*, for MB released from PLA/PEG blends.

Sample	*Q*	*k_1_*, d^−1^	*k_2_* × 10^−2^, d^−1/2^	*R* ^2^
PLA/PEG/MB H10	0.55	0.92	1.55	0.99
PLA/PEG/MB U3min-50W	0.52	1.61	1.30	0.99
PLA/PEG/MB U3min-150W	0.75	2.15	1.04	0.99
PLA/PEG/MB U3min-250W	0.64	1.44	1.32	0.99

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
