# Peer review of "Synergistic Effect of Ultrasound and Polyethylene Glycol on the Mechanism of the Controlled Drug Release from Polylactide Matrices"

_polymers, 2019, doi:10.3390/polym11050880_

Round 1

Reviewer 1 Report

This manuscript prepared a polymeric tablet (drug delivery system) comprising degradable PLA and hydrophilic PEG with encapsulated model drug (methylene blue) prepared by melt processing. The effect of ultrasound and PEG contents on the drug releasing behavior has been investigated based on the diffusion-controlled process. These studys and analysis are good which highly explained the proposed strategy. However,  the author should clearly explain the following points to emphasize the scientific or practical impact before the current manuscript being processed to publication.

1.    How will the ultrasound be applied for in vivo drug releasing using the current method?

2.    Please explain which drug has similar physical property as MB (small water-soluble molecule). Otherwise, using MB as model drug will be meaningless.

3.    How is the stability of PEG during melt blending at 170C?

4.    What’s the merit of this study and what problem could be solved through the current study? PLA and PEG are not new materials, and the releasing strategy has neither scientific impact nor practical application potential.

5.    Please explain in details on the tablets, e.g., size, shape, surface morphology, which are important for practical application.

6.    Why do you need study 120 day’s stability test? What kind of drug need to be released for 120 days and what specific issue could be worked out?

Author Response

1.      How will the ultrasound be applied for in vivo drug releasing using the current method?

Thank you for your suggestion. Ultrasound may be applied to improve or control the distribution of the drug in the preparation and it could also be used to make some drug delivery system with special structure by controlling the polymer phase structure.

2.      Please explain which drug has similar physical property as MB (small water-soluble molecule). Otherwise, using MB as model drug will be meaningless.

Thank you for your suggestion. It was reported in some literatures (A. J. Chung, and M. F. Rubner, Methods of Loading and Releasing Low Molecular Weight Cationic Molecules in Weak Polyelectrolyte Multilayer Films, Langmuir, 2002, 18 (4): 1176–1183.;Anthony R. Disanto, John G. Wagner, Pharmacokinetics of Highly Ionized Drugs II: Methylene Blue—Absorption, Metabolism, and Excretion in Man and Dog after Oral Administration, Journal of Pharmaceutical Sciences, 1972, 61(7):1086-1090.) that MB was used in the pharmacokinetics or used as the indicator to observe the release behavior, so we used MB as the model drug to observe the release behavior

3.       How is the stability of PEG during melt blending at 170C?

Thank you for your suggestion. PEG during melt blending at 170 was stable as the previous literature reported, (Thongpina C, Tippuwanan C, Buaksuntear K, et al. Mechanical and Thermal Properties of PLA Melt Blended with High Molecular Weight PEG Modified with Peroxide and Organo-Clay, Key Engineering Materials. Trans Tech Publications, 2017, 751: 337-343.)

4.    What’s the merit of this study and what problem could be solved through the current study? PLA and PEG are not new materials, and the releasing strategy has neither scientific impact nor practical application potential.

Thank you for your suggestion. Most of laboratory investigations of polyester biomaterials rely on additives and solvent-casting manufacturing techniques which do not allow translation into industrial processing because of the present of the volatile organic compounds.

5.      Please explain in details on the tablets, e.g., size, shape, surface morphology, which are important for practical application.

Thank you for your suggestion. In our experiment, the tablets were with the diameter of 10mm and the thickness of 0.4mm.

6.    Why do you need study 120 day’s stability test? What kind of drug need to be released for 120 days and what specific issue could be worked out?

Thank you for your suggestion. In our study, the drug release system was designed for long term drug release, so we do the in-vitro release for 120 days.

Reviewer 2 Report

It is a novel strategy to introduce ultrasound to polymer melt. It can help to decrease the phase size, especially for immiscible polymers. It would be better to examine the phase distribution with and without ultrasound treatment. When the ratio of PLA and PEG is 50:50, an interpenetrating polymer network (IPN) should be considered for the drug release.

Author Response

It is a novel strategy to introduce ultrasound to polymer melt. It can help to decrease the phase size, especially for immiscible polymers. It would be better to examine the phase distribution with and without ultrasound treatment. When the ratio of PLA and PEG is 50:50, an interpenetrating polymer network (IPN) should be considered for the drug release.

Thanks very much for your comments and we will consider the interpenetrating polymer network (IPN) in our future research. 

Reviewer 3 Report

This work presents the new synthetic approach integrating ultrasound and melting extrusion for PLA/PEG. The drug release behaviors of the composite material using MB as a model drug was characterized for 120 days. The effect of the ultrasound step was investigated. The work is quite systematic, however, I did not see very sound scientific significance in the work. The aim of the work is not clear either, I would suggest major revision before acceptance for the journal.  Several questions:

1, Figure 1 shows the release curves of MB from PLA as a function of power and length of the ultrasound treatment. First of all, the ultrasound treatment times are the same for all three samples with the ultrasound treatment.  There is no time differences, so there is no function of length of ltrasound treatment. Secondly, the ultrasound power showed effect on the drug release behavors, 150 W showed highest release amount and then 50W, the 250W showed lowest drug release amount, even lower than the PLA without ultrasound. The discussion did not mentioned the power effect and did not explain why 150W show highest release amount.

2, In Figure 3, the effect of PEG content on the release behavior was shown. PLA-50%PEG sample presented very obvious burst release in the first few days as well as the other PEG blended samples. Authors should specify at the beginning what the objective of this composite materials for, long term drug release or short term in terms of applications, each one should have its specific drug release requirement.

3, The experiment design also should show the weight loss at different time points, especially show the weight loss data at very first few days, to identify if the release has been completed after the burst release, if so, there is no point to conduct the 120 days experiments.

4, In Figure 5, the ultrasound effect on the release behaviors of MB from PLA-35%PEG is not consistent with the Figure 2. Please explain why. The burst release are also very obvious, the weight loss for the composite materials should be presented.    

5, The fitting in Figure 8 does not make sense to me, suggesting the equation might not the perfect one. Please do a better fitting.  

Author Response

1. Figure 1 shows the release curves of MB from PLA as a function of power and length of the ultrasound treatment. First of all, the ultrasound treatment times are the same for all three samples with the ultrasound treatment.  There is no time differences, so there is no function of length of ultrasound treatment. Secondly, the ultrasound power showed effect on the drug release behaviors, 150 W showed highest release amount and then 50W, the 250W showed lowest drug release amount, even lower than the PLA without ultrasound. The discussion did not mentioned the power effect and did not explain why 150W show highest release amount.

Thank you for your suggestion. It was reported the power effect as “From about 30 days to 120 days, when the ultrasonic power was 50W or 250W, the cumulative releases of MB were less than that from PLA without ultrasound treatment. However, when the ultrasonic power was 150 W, both the release rate and the cumulative release of MB from PLA were improved, with the amount being longer then both the non-treated PLA and the 50W and 250W treated samples for the entire period of measurement.” Then we discuss this phenomenon and explain it as We suggest the main reason for this phenomenon was that the ultrasound had two different kinds of effects on the release of MB from PLA: firstly, ultrasound would lead to the degradation of the PLA matrix resulting in more PLA chains with lower molecular weight in the tablets[36], there would be more hydroxyl group, so the degradation and hydrophilicity of PLA would be improved and this kind of effect would enhance the release of the MB; secondly, ultrasound would improve the dispersion of MB in the PLA matrix, there was more MB dispersed in the internal part of the tablet, which would take longer to diffuse into the medium, so this effect would decrease the release of MB. These two effects competed with each other and led to the final results. In the first stage of the immersion, the degradation effect of ultrasound was predominant; after 20 days immersion, the effect of the polymer matrix on the release became weak and the distribution and diffusion of the drug was the dominant effect. We assume that when the ultrasound was applied at 150 W for 3 minutes, these two effects achieved a balance, so the molecular weight of PLA was decreased obviously and the dispersion of the drug MB was improved slightly, which led to the increase of both the release rate and cumulative release of MB for the entire measurement period.”

2. In Figure 3, the effect of PEG content on the release behavior was shown. PLA-50%PEG sample presented very obvious burst release in the first few days as well as the other PEG blended samples. Authors should specify at the beginning what the objective of this composite materials for, long term drug release or short term in terms of applications, each one should have its specific drug release requirement.

Thank you very much. In our research, this drug delivery system was designed for long term drug release, so the very obvious burst release was unexpected .

3. The experiment design also should show the weight loss at different time points, especially show the weight loss data at very first few days, to identify if the release has been completed after the burst release, if so, there is no point to conduct the 120 days experiments.

Thank you very much. We have already consider to measure the weight loss at different time points, however, the samples were required to be dried to a constant weight in a vacuum oven before weighing, this may stop the in-vitro release study, so we didn’t measure the weight loss at different time points. Since MB was blue and shows very good solubility in water, it was very easy for us to observe the change of the color of the tablets and we could roughly identify if the release has been completed after the burst release

4. In Figure 5, the ultrasound effect on the release behaviors of MB from PLA-35%PEG is not consistent with the Figure 2. Please explain why. The burst release are also very obvious, the weight loss for the composite materials should be presented.   

Thank you very much for the comments. In Figure 2, the release behaviors of MB from pure PLA was affected by ultrasound which could lead to the degradation of the PLA matrix and improved the distribution of MB in the PLA. In Figure 5, when PEG was added, the carrier of the MB in the tablet became the PLA/PEG blends instead of PLA, and PLA/PEG blends were degraded by the chemical effect of ultrasound; this led to a faster hydrolysis degradation rate of the PLA/PEG matrix and a faster release rate of MB. At the same time, the physical effect of ultrasound might improve the dispersion of PEG and MB in PLA. The better dispersion would have two results: one was that the more and better dispersed release channels, which were formed by the erosion of PEG, would increase the release of the drug. The other was that the better dispersion of MB meant relatively more MB dispersed in the PLA phase, so the drug release became more difficult. When the ultrasound was applied at 150 W for 3 minutes, these two kinds of results achieved balance, so the release of MB from the tablets was the fastest and the most. So the ultrasound effect on the release behaviors of MB from PLA-35%PEG is not consistent with the Figure 2.

5. The fitting in Figure 8 does not make sense to me, suggesting the equation might not the perfect one. Please do a better fitting. 

Thank you very much. We do a better fitting and update the data and figure 8 in the manuscript. 

Reviewer 4 Report

 The authors have investigated PEG 6K additives and ultrasonic post-treatment of PLA films for long term release of a model drug—methylene blue.  The effects of PEG 6K are predictable and has been reported many times.  The effects of post-treatment with ultrasound in the melt phase has not been sufficiently investigated in the literature and could have high industrial applicability.  However, there are a number of assumptions that are made that are not supported by the data.  Also, discussion of their work compared to recent literature is weak. The manuscript should be published after the authors have addressed the following points:

1.             The data on ultrasound drug release displays no correlations with time of application.  The authors justify this by two mechanism: 1)”firstly, ultrasound would lead to the degradation of the PLA matrix resulting in more PLA chains with lower molecular weight in the tablets[36]”, 2) “ultrasound would improve the dispersion of MB in the PLA matrix.”  The authors need to support these statements with supporting data.  How does the molar mass of PLA degrease with 50-250 watts?  What is the surface/core dispersion of MB before and after ultrasound?

2.             PLA and PEG molar mass and polydispersity need to be included within the methods for other to reproduce the data.

3.             Figure 4 and 6 are not correctly labelled—there is no way to determine which ones are the cross-section and which is the surface. The legend state a and a1—but these are not present in the photos.

4.             Discussion incorporating methods to increase drug release with other post-extrusion methods, such as plasma treatment or irradiation.  Examples:

https://doi.org/10.1016/j.jconrel.2017.09.023

https://pubs.acs.org/doi/abs/10.1021/am500454b

Radiat Phys Chem. 2015;109:73-82.

https://doi.org/10.1080/00914037.2017.1309543

5.             Discussion incorporating methods to increase drug release via the hot-melt extrusion parameters:

https://doi.org/10.1016/j.xphs.2019.02.024

Ocular implant made by a double extrusion process. 2011. U.S. Patent 8,034,370,

Author Response

1. The data on ultrasound drug release displays no correlations with time of application.  The authors justify this by two mechanism: 1)”firstly, ultrasound would lead to the degradation of the PLA matrix resulting in more PLA chains with lower molecular weight in the tablets[36]”, 2) “ultrasound would improve the dispersion of MB in the PLA matrix.”  The authors need to support these statements with supporting data.  How does the molar mass of PLA degrease with 50-250 watts?  What is the surface/core dispersion of MB before and after ultrasound? Thank you very much. In our previous research, we have found that ultrasound could lead to the degradation of the PLA, which was reported in “W. Bao; H. Wu; S. Guo; A. Paradkar; A. Kelly; E. Brown; P. Coates, Effect of Ultrasound on Molecular Structure Development of Polylactide. Polym.-Plast. Technol. Eng., 2014, 53, 927-934.” The particle size of MB becomes smaller after ultrasonic treatment, and the dispersion in PLA is more uniform as shown in the Figure. It can be observed that the color of the sample becomes darker with the increase of the time of the ultrasound treatment.  This means the ultrasonic oscillation can enhance the dispersion of MB in PLA. 2.  PLA and PEG molar mass and polydispersity need to be included within the methods for other to reproduce the data. Thank you very much. In our manuscript, we make a supplement “The PLA (REVODE101, Mn=5.3×104, Mw/Mn=1.935) used in the experiment was provided by Zhejiang Hisun Biomaterials Co., Ltd. China.” “The PEG with molecular weight of 6000 used was provided by Tianjin Bodi Chemical Co., Ltd. China.” 3. Figure 4 and 6 are not correctly labelled—there is no way to determine which ones are the cross-section and which is the surface. The legend state a and a1—but these are not present in the photos. Thank you very much. We have re-labelled the figure 4 and 6. 4. Discussion incorporating methods to increase drug release with other post-extrusion methods, such as plasma treatment or irradiation.  Examples: https://doi.org/10.1016/j.jconrel.2017.09.023 https://pubs.acs.org/doi/abs/10.1021/am500454b Radiat Phys Chem. 2015;109:73-82. https://doi.org/10.1080/00914037.2017.1309543. Thank you very much. In our manuscript, we studied the synergistic effect of ultrasound and polyethylene glycol on the drug release behavior. Ultrasound is different from the plasma treatment or irradiation which mainly shows chemical effect. It was reported that ultrasound was applied to assist melt processing and it showed both physical and chemical effects on the polymer and its blends and composites. We want to change the structure of the drug carrier by both physical and chemical effects of ultrasound.(Zhao, L.; Li, J.; Guo, S.; Du, Q. Ultrasonic oscillations induced morphology and property development of polypropylene/montmorillonite nanocomposites. Polymer 2006, 47 (7), 2460-2469. Wu, H.; Guo, S. Improved properties of metallocene-catalyzed linear low density polyethylene/polypropylene blends during ultrasonic extrusion Chin. J. Polym. Sci. 2007, 25 (4), 357–364.) 5.  Discussion incorporating methods to increase drug release via the hot-melt extrusion parameters: https://doi.org/10.1016/j.xphs.2019.02.024 Ocular implant made by a double extrusion process. 2011. U.S. Patent 8,034,370, Thank you very much. The hot-melt extrusion was widely used in pharmaceutical research and the effect of hot-melt extrusion process parameter has been reported. (Maniruzzaman, M., Boateng, J. S., Snowden, M. J., & Douroumis, D. (2012). A Review of Hot-Melt Extrusion: Process Technology to Pharmaceutical Products. ISRN Pharmaceutics, 2012, 1–9.) In our research, we focused on the effect of ultrasound on the release behavior of the drug delivery system and we only use HME to mix the polymer materials and model drug.

Round 2

Reviewer 1 Report

accept

Reviewer 2 Report

It was a good work.

Reviewer 3 Report

The revised version is good to be accepted.